# Towards an objective theory of subjective liking: A first step in understanding the sense of beauty

S. Mazzacane[1]*, M. Coccagna[1], F. Manzella[2], G. Pagliarini[2], V. A. Sironi[3], A. Gatti[4], E. Caselli[1], G. Sciavicco[2]

**1** CIAS Interdepartmental Research Center (Dept. of Architecture, Dept. of Chemical, Pharmaceutical and Agricultural Sciences), University of Ferrara, Ferrara, Italy, **2** Dept. of Mathematics and Computer Science, University of Ferrara, Ferrara, Italy, **3** CESPEB Research Center, Neuroaesthetic Laboratory, University Bicocca, Milan, Italy, **4** Dept. of Humanistic Studies, University of Ferrara, Ferrara, Italy

* guido.sciavicco@unife.it

**Data Availability Statement:** All relevant data for this study are publicly available from the Kaggle repository (http://doi.org/10.34740/kaggle/dsv/5045028).

## Abstract

The study of the electroencephalogram signals recorded from subjects during an experience is a way to understand the brain processes that underlie their physical and emotional involvement. Such signals have the form of time series, and their analysis could benefit from applying techniques that are specific to this kind of data. Neuroaesthetics, as defined by Zeki in 1999, is the scientific approach to the study of aesthetic perceptions of art, music, or any other experience that can give rise to aesthetic judgments, such as liking or disliking a painting. Starting from a proprietary dataset of 248 trials from 16 subjects exposed to art paintings, using a real ecological context, this paper analyses the application of a novel symbolic machine learning technique, specifically designed to extract information from unstructured data and to express it in form of logical rules. Our purpose is to extract qualitative and quantitative logical rules, to relate the voltage at specific frequencies and in specific electrodes, and that, within the limits of the experiment, may help to understand the brain process that drives liking or disliking experiences in human subjects.

## Introduction

*Neuroaesthetics* was defined by Zeki [1] as the scientific approach to the study of the perception of beauty, that is, the study of *liking* (to be distinguished from *wanting*, as in [2]) in a broad sense. The approaches to neuroaesthetics vary very much in the recent scientific literature, but they can be fundamentally divided into *top-down* processes, in which the essence of beauty undergoes an axiomatic treatment in which the subjective feeling is broken down to its constituting elements, and *bottom-up* ones, in which some kind of objective data is analysed and related to the subjective expression of beauty. While the former is certainly fascinating from an epistemological point of view (it tries to answer the question of whether *beauty can be defined*), and it is, in essence, a philosophical exercise, the latter is a neurophysiological one, based on real data, systematic, and carried on with modern analysis techniques. Such an approach had already been suggested as early as the second half of the 19th century (the so-

**Funding:** The author(s) received no specific funding for this work.

**Competing interests:** The authors have declared that no competing interests exist.

called Weber-Fechner Law), as an attempt to create a mathematical relationship between stimulus and perception, although not to quantify beauty, and it is largely the most common one.

Bottom-up strategies aimed to understand the mechanisms of the brain can be broadly classified into (functional) MRI (*magnetic resonance*) image processing and interpretation, and EEG (*electroencephalogram*) signal processing; in some cases the approaches are combined, or other brain analysis techniques are adopted. Even focusing on EEG only, the relevant work is huge. It includes, among other elements: experiments aimed to understand human emotions (of which liking is just one example, often not considered as a single emotion but as a combination of several ones [3]); experiments designed with own or public datasets; case studies in laboratory conditions or real-life situations; approaches focused on qualitative or quantitative descriptions, in some cases time-related; experiments in which the stimuli vary from images, to audio, to movies, sometimes computer-generated. Performing a complete review of such a body of work is beyond the scope of this paper; just the most recent review paper of this field [4] surveys as much as 102 contributions selected from a pool of 654 potential ones. Such an high interest in the subject is also witnessed by the availability of several public datasets, which include SEED, DEAP, MANHOB-HCI, DREAMER, AMIGOS, DECAF, INTERFACES, among many others (see [5] for a complete list of public datasets). From 1985 to 2020, several authors have worked, in particular, on the problem of classifying emotions from EEG signals using statistical/machine learning techniques with approaches that are comparable with ours. To begin with, Ray and Cole [6] approached the problem of examining the effect of attentional demand during cognitive and emotional tasks, in a pair of experiments with 18 and 40 subjects, respectively; the experiments were carried out in a clinical context and, in terms of emotions, the focus was on distinguishing between happiness and sadness. In [7] the authors focused on specific features extracted from 10-electrode EEGs to be correlated with anger, sadness, joy, and relaxation; their experiment involved 5 subjects. Schmidt and Trainor [8] found that the pattern of asymmetrical frontal EEG activity distinguished valence of the musical excerpts, in an experiment with 59 subjects, 4-electrode EEG, and music-induced stimulation. In [9] the authors discussed how the changes in the electrical activity of the human brain related to distinct emotions, in an experiment with 6 subjects and 63-electrode EEG, focusing on anger, disgust, fear, happiness, sadness, and surprise, elicited via the exposition of the subjects to movie clips; three of the same authors later focused on signal preprocessing, and designed a classifier for 5 emotions, that is, disgust, happy, surprise, fear and neutral, from an experiment with 20 subjects [10]. Only two emotions, instead, were the subject of an experiment by Li and Lu [11], carried out on 10 subjects, stimulated, in a clinical context, by being exposed to pictures displaying facial expressions; while in this case very high accuracies were reached, the total number of trials was relatively low. Schaaff and Schultz [12] performed an experiment with 5 subjects, 4-electrode EEG, with the aim of building a classifier to recognize positive, neutral, and negative emotions in humans, with relatively low success. In [13] the authors studied EEG correlates on emotions using features extracted by Gaussian mixtures of EEG spectrograms; their experiment involved 31 subjects, 8-electrode EEG, two emotions (positive and negative), and two arousal states (calm and excited); in [14], in a similar experiment with 26 subjects, the performance of a EEG-based emotion recognition system based on a self-organizing map to identify the boundaries between separable regions were studied. Petrantonakis and Hadjileontiadis [15] proposed a novel emotion evocation and EEG-based feature extraction technique, combined with higher order crossings analysis, for feature extraction and robust classification; their experiment involved 16 subjects, and 3-channel EEG. In [16], instead, the authors propose a methodology for robust classification of EEG biosignals into four emotional states; their experiment involved 28 subjects and 19-electrode EEG, and emotions were elicited with pictures observation. Liu and Sourina [17, 18] worked

on a real-time fractal dimension based valence level recognition algorithm from EEG signals, which allows a sort of continuous classification of up to 16 different emotions; their work is based on a public dataset. Nie, Wang, Shi, and Lu [19] considered the problem of classifying positive and negative emotions from 62-electrode EEG, stimulated by the action of watching movie clips from relatively famous, and emotional movies; the high accuracies reached in this case by using support vector machines is to be considered in the perspective of a very reduced number of subject (3) and trials (12 per subject). Later on, the same authors considered, the problem of classifying several types of emotions including amusement, anger, contentment, disgust, fear, neutral, sadness, and surprise [20], using, again, support vector machines; their experiment involved 6 subjects, and movies were, once more, used to generate emotions in laboratory conditions. In [21], Hadjidimitriou and Hadjileontiadis used different feature extraction approaches and classifiers, and focused on the discrimination between subjects' EEG responses to self-assessed liked or disliked music, in an experiment with 9 subjects and 14-channel EEG. Jenke, Peer, and Buss proposed a statistical method to select electrodes and features that separate classes well in the problem of emotion classification; their experiment was carried on 16 subjects with a 64-channel EEG [22]; at the same conference, Rozgić, Vitala-devuni, and Prasad addressed single-trial binary classification of emotion dimensions, using a public EEG dataset [23]. In [24], the authors proposed a deep learning method is proposed to recognize emotion from raw EEG signals using LSTM neural networks and a public dataset. Neural networks were also applied to the same problem in [25], with several public and private datasets, and also in [26], again on a public dataset, as well as in [27].

Several considerations can be drawn from reviewing the literature. First, most of the existing work on computational neuroaesthetics focuses on artificial, 2D images and 3D shapes designed with experimental purposes, or movies, and experiments are almost never carried on in real contexts. Moreover, EEG signals have been analysed with functional machine learning techniques that do not allow, in general, the interpretation of the resulting models, nor the extraction of explicit rules, and can only be judged by their statistical performances. The most common approaches vary from univariate/multivariate inferential and descriptive statistical approaches, to support vector machines, to neural networks; in some other cases, EEG signals have been classified with other methods, e.g., hidden Markov models, but not in the case of emotions. Furthermore, even if liking is one of the labels explicitly present in public datasets (e.g. DEAP), it has been rarely singled out in the experiments; quickly surveying among the authors that do mention it [23, 24, 26, 28], we may observe that the average accuracy of classification ranges from 60% to 86%, although the peculiarities of the single experiments and their conditions hardly allow for a definitive comparison. Finally, most of the effort has been directed to build complete classification models, and only in a few cases to understand, in a systematic fashion, the role of each electrode placement, frequency, and feature in relation with the class to be learned. In this paper, we consider data collected during a real-world experiment, analysing the explicit and implicit reactions of participants, using different kinds of sensors, during their visit to an art exhibition. Our data are part of a larger biosignals dataset recorded to evaluate the participants' reactions during the observation of paintings, which included ECG (*electrocardiogram*), EDA (*electrodermal activity*), two different tools for EEG recording, the *gaze pattern*, the participants' main characteristics (age, gender, education, familiarity with art, etc.) and their explicit judgments about paintings. Focusing on the EEG signal only, in this paper we consider the problem of extracting explicit rules that underly the electric signals at specific frequencies of specific electrodes, using a new techniques for extraction of knowledge from time series, based on *temporal decision trees* and *temporal random forests* [29, 30]. In particular, we consider the problem of identifying the most relevant sub-group of electrodes whose signal correlates to the subjective liking experience, the most relevant sub-

group of frequencies at which the sensation seems to reveal itself, and the most relevant sub-group of feature functions that allow one to highlight such information, with the aim of extracting explicit, logical rules that relate time (qualitatively) and power (quantitatively) during the experience. We have chosen to focus on the available EEG data only, and, among them, those gathered with a single type of equipment. While other biosignals, such as EDA, did not seem to contain relevant information for the task at hand, this is not the case, for example, for gaze patterns; however, treating gaze data is essentially different in both techniques and approaches, and requires a dedicated effort.

Our approach is not only novel in this particular area, but also in medical and biosignal treatment in general, and it allows one to analyse very large quantities of data. Moreover, it should be observed that the fluctuation between a philosophical, artistic, neurological, techno-logical or computational viewpoints, just to name a few of the most important one when base-lining this project, gave us the opportunity to discuss the results taking into account different knowledge; such an interdisciplinary approach, working with a diverse team right from the hypothesis stage, contributed to exploit insights and results acquired in disciplinary fields that would have otherwise remained isolated.

## Data origin

The dataset used in this paper has been collected during the exhibition *Painting affections: sacred painting in Ferrara between the '500 and the '700*, set up at the Estense Castle in Ferrara (Italy) from the 26th of January to the 26th of December 2019 [31, 32]. The original exhibition included 54 paintings, of which 18 have selected according to some exclusion criteria, to avoid the risk that people may be distracted when approaching to the observation point (too small paintings), by details unrelated to the artwork itself (damaged paintings) or by the difficulties in focusing on single paintings (paintings that are part of a series). The experiment included two different ways to collect EEG data, to analyse benefit and boundaries of different tools, procedures and final results. However, in this paper we analyse only the outcomes coming from subjects wearing a dry electrode EEG cap *Waveguard™touch* by *ANT Neuro* in the 64-channel variant during the exhibition. To minimise the influence of movements, the 16 people involved in this particular setting attended to the trial while using a wheelchair pushed by a researcher. Each experiment produced a single long recording per subject, ranging from the beginning to the end of the exhibition; therefore all data required a preliminary slicing operation, based on the actual time at which the subject was standing in front of each painting, in order to be able to analyse the data related to each specific painting. To better take into account the strictly personal reactions of people to emotions and the environment, each subject has been exposed to a blank image (a white wall) for 60 seconds before his/her trial, which consisted of a 60-seconds observation of each of the 18 paintings. Being recorded at a sampling rate of 512 Hz, each data slice consisted of a time series of 30720 points. Each subject was asked to express an integer in the range [0, 50] as an expression of their subjective liking score of the painting, using a *Visual Analog Scale* (*VAS*) scale [33]. Since not all subject-painting pairs were recorded, the resulting dataset is composed of 248 instances, each instance being described by 64 time series (one for each electrode) of 30720 points and labelled by the relative liking score.

Statistically-wise, subjects were analysed by their gender, age, nationality, education level, and subjective interest in the painting arts. The subjects' set is composed of 10 males and 6 females with an average age of 32.19±15.81, with a subjective interest in the painting arts of 25.93±7.58 on a scale of [0, 50] (see Fig 1, top). Moreover, 14 subjects (that is, the 87.5%) were Italian, 1 (6.25%) was Portuguese and 1 (6.25%) was Russian, and the education level was dis-tributed as follows: in the 62.5% (10 subjects) of the cases, the highest level of education was

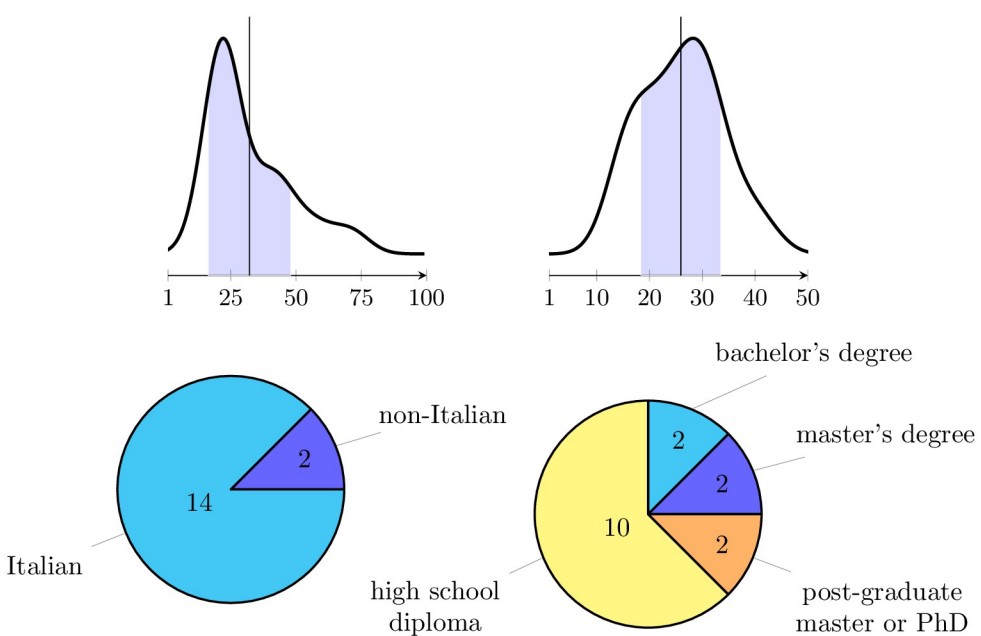

**Fig 1. Distributions of age (top-left), interest in painting arts (top-right), nationality (bottom-left) and education level (bottom-right) of the subjects.**

high school diploma, in the 12.5% (2 subjects) was bachelor's degree, in the 12.5% (2 subjects) it was master's degree, and in the remaining 12.5% (2 subjects) cases it was PhD, or other type of postgraduate (see Fig 1, bottom). In terms of sample size, and other relevant hypothesis on the distribution of data, no pre-qualifying tests (e.g., power analysis) were performed, as they are generally unnecessary when data are analyzed with machine learning techniques; this is particularly relevant in observational studies that require complex preparation and whose sample size and distribution cannot be easily controlled at design time.

## Ethic statement

This study was approved on the 14th of April, 2019, by the Ethics Committee of "Area vasta Emilia Centro—Regione Emilia Romagna", (University of Bologna), with reference number 282/2019/Sper/UniFe, and registered on ISRCTN with number 70216542. Data collection has been performed according to a strict protocol approved by the committee, specifically concerning informed consents and privacy. The *NEVArt* website [34] and a set of downloadable documents explained purpose and rules before people completed the participation form. In particular, we prepared a short summary of the research purposes, development, data processing, as well as a copy of the informed consent, that have been sent by email to participants when confirming data and time of the experiment and the instruction to participate. The final informed consent has been signed just before the trial, in paper, to be able to give any further detail, if needed, before the experiment (and a copy was returned to the participant). The informed consent (as well the database and the data processing description) has been provided both in Italian and English language. The research database includes only adults; a small number of minors have been tested (6 people, between 11 and 17), with a specific written parent approval, but not included in the dataset. These trials have been provided to give the chance to kids (3 subjects) to participate with their family group, as well as to our, still minor, trainees from secondary school (3 subjects).

## Preprocessing

The raw EEG signals (see an example in Fig 2), originally sampled at 512 Hz were, first, re-sampled at 104 Hz in order to lower the Nyquist frequency to 52 Hz [35] while removing possible noisy frequency bands in the high range, and, then, processed with a Short Time Fourier Transform (STFT), in order to retain both the frequency and time domains. The (0, 52] Hz spectrum (0 excluded, 52 included) was divided into 13 equally wide bands of frequencies ($F_1$, ..., $F_{13}$), with a resulting width of 4 Hz, as per the result of the initial screening. The STFT time window size was set to 50 *ms* with a step time of 20 *ms*, and the resulting time series per each trial were 3379 points long. For reference and comparison with existing work, such 13 bands can be grouped into the five standard relevant wave patterns (see Fig 3 for an example): $\delta$, ranging in $0 - 4$ Hz (usually associated with slow-wave sleep), $\theta$, ranging in $4 - 8$ Hz (usually associated with phase 1 and 2 of non-REM sleep and with REM sleep), $\alpha$, ranging in $8 - 12$ Hz (usually associated with waking state with closed eyes and instants prior to fall asleep), $\beta$, ranging in $12 - 32$ Hz (usually associated with intense mental activity), and $\gamma$, ranging in $32 - 52$ Hz (usually associated with states of particular stress). Observe that each $F_i$ ($1 \leq i \leq 13$) is obviously finer than a normal band $b \in \{\alpha, \beta, \gamma, \delta\}$; the resulting dataset is clearly richer than those obtained in more classic approaches (e.g., using the four bands directly), but not so rich to generate learning noise (e.g., using 1 Hz bands). This makes it possible to extract more statistically reliable models at the expenses of a more computationally demanding extraction phase. Our information extraction approach being not distribution-dependent and based on simple variance renders the typical eye blinking and movement noise cancellation unnecessary [36], effectively simplifying the whole procedure; moreover, in this way, we expect to extract a model that is, at least in principle, robust to eye blinking and neck muscle contraction noise, to be used in a real environment.

The scores among the dataset, in the range [0, 50], were non-normally distributed ($p = 10^{-56}$, with a Kolmogorov-Smirnov normality test). For the purpose of learning, we treated the resulting problem as a classification one, by discretizing the liking scores into three categories (dislike, neutral, like). For this *data binning* step, the three different pairs of thresholds $\langle < 25, \geq 25 \rangle$, $\langle < 17, \geq 34 \rangle$, and $\langle < 10, \geq 41 \rangle$ were considered, giving rise to three datasets ($\mathcal{D}_{25}, \mathcal{D}_{34}, \mathcal{D}_{41}$) encompassing, respectively, 248 instances (125 dislike versus 123 like), 167 instances (86 dislike versus 81 like), and 91 instances (46 dislike versus 45 like). Binning was equal-length despite the distribution not being normal, as it still shows a symmetrical behaviour with respect to the median value, as shown in Fig 4.

In order to prepare the datasets for the learning phase, it is necessary to further treat the signals and drastically reduce the number of points and electrodes that are effectively used. While the problem of attribute/feature selection is widely known in the literature, treating temporal

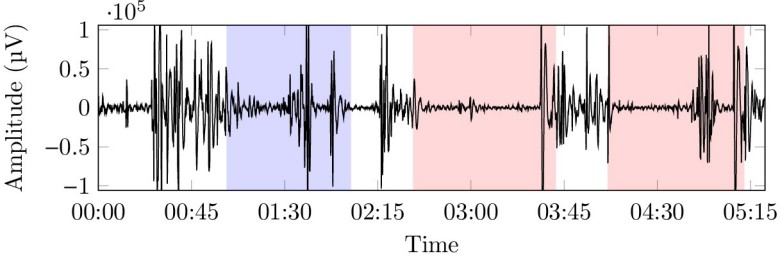

**Fig 2. Example of a raw EEG signal of a single subject, from a single electrode; the first highlighted area corresponds to the blank portion of the observation, while the others correspond, each, to the observation of different paintings (here, only two paintings are shown in the picture).**

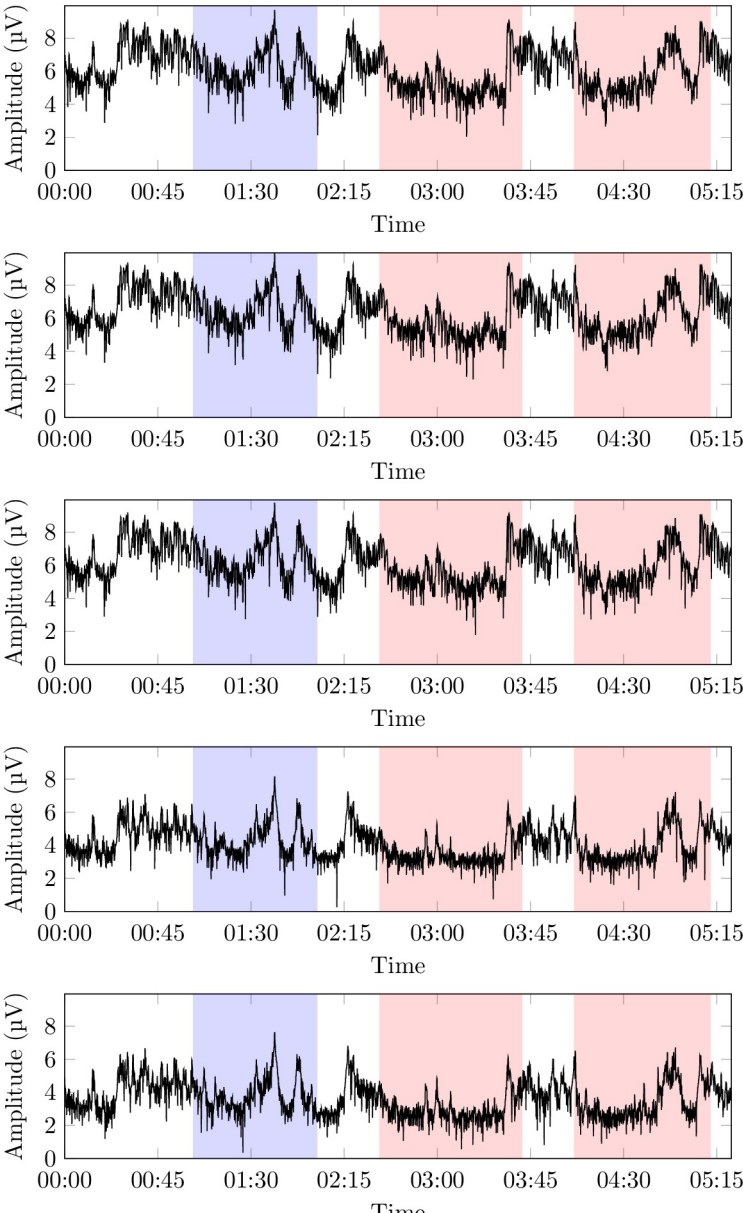

**Fig 3.** From top to bottom, the intensity of voltage at the band $\delta$, $\theta$, $\alpha$, $\beta$ (specifically, the interval of $\beta$ included in $F_6$), and $\gamma$ bands (specifically, the interval of $\gamma$ included in $F_{11}$) during the trial of example in Fig 2, after preprocessing. All graphics are $\log_{10}$-normalized. As before, the first highlighted area corresponds to the blank portion of the observation, while the others correspond, each, to the observation of different paintings (again, only two paintings are shown in the picture).

features is not very common. We modelled this classification problem as a *multivariate temporal series* classification problem. Naïve symbolic treatment of time series consists of simple feature extraction for further analysis; however, as suggested in [37, 38], elementary features can be combined with more elaborate ones. Prior to the knowledge extraction phase, therefore, we aimed to highlight which electrodes and which measures (i.e., feature functions) are more prone to have a role in this problem. To this end, we performed a statistical analysis based on

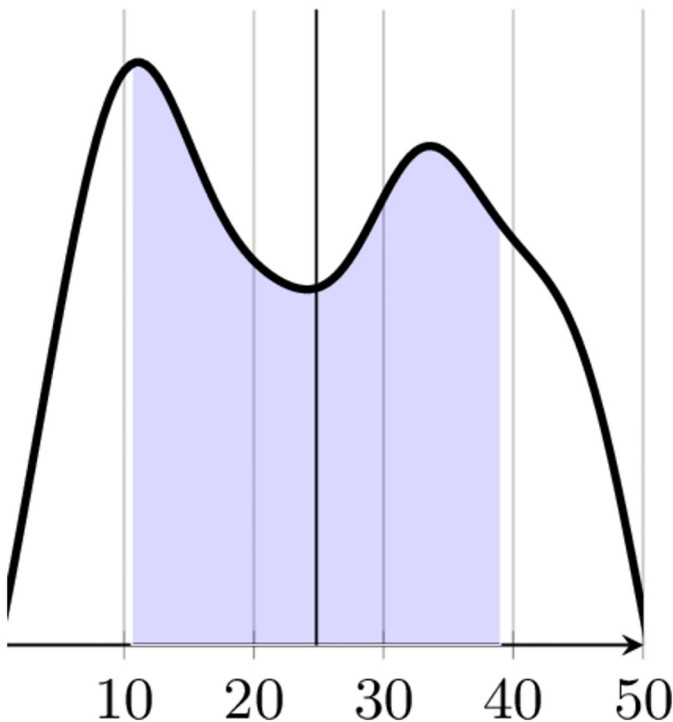

**Fig 4. Distribution of liking scores.**

the assumption that, when coupled with a given measure, a frequency is more informative if it displays a higher variance across the instances. In the following, we shall refer to *feature* as the triple *electrode-frequency (band)-measure*, whereas, in particular, a measure is a specific function applied to a time series (or to an interval of a time series). Among all possible measures we considered 22 functions generically considered as informative for time series, listed in [37], plus *minimum, maximum*, and *average*, for a total of 25; they are all listed in Table 1. For each dataset ($\mathcal{D}_{25}$, $\mathcal{D}_{34}$, or $\mathcal{D}_{41}$), we proceeded as follows:

- We applied all measures to the signal at all frequencies from all electrodes, obtaining $13 \times 64 \times 25 = 20800$ (potential) features—in this way a dataset ends up being described, therefore, by 20800 temporal features.

- We computed the (min-max normalized) variance of each feature across the whole dataset.

- We listed the best electrodes in non-decreasing order of their *e-score*, that is, the average normalized variance across all frequencies and all measures.

- We selected three sub-groups of electrodes from the best ones (w.r.t. their e-score), of, respectively, 1, 5, and 10 electrodes each.

- For each given group, we listed the best measures in non-decreasing order of their *m-score*, that is, the average normalized variance across all (selected) electrodes and all frequencies.

The result of this process is a selection of pairs electrode-measure that score well across all frequencies bands, and it is visible in Figs 5–7. In all figures, the right-hand side shows the

**Table 1. 25 statistical measures for time series (including 22 measures from [37]).**

| measure | symbol |
| --- | --- |
| minimum | MIN |
| maximum | MAX |
| average | AVG |
| mode of $z$-scored distribution (5-bin histogram) | Z5 |
| mode of $z$-scored distribution (10-bin histogram) | Z10 |
| longest period of consecutive values above the mean | C |
| time intervals between successive extreme events above the mean | A |
| time intervals between successive extreme events below the mean | B |
| first $1/e$ crossing of autocorrelation function | FC |
| first minimum of autocorrelation function | FM |
| tot. power in lowest 1/5 of frequencies in the Fourier power spectrum | TP |
| centroid of the Fourier power spectrum | CE |
| mean error from rolling 3-sample mean forecasting | ME |
| time-reversibility statistic $\langle (x_{t+1-x_t})^3 \rangle t$ | TR |
| automutual information ($m = 2$, $\tau = 5$) | AI |
| first minimum of the automutual information function | FMAI |
| proportion of successive differences exceeding $0.04\sigma$ | PD |
| longest period of successive incremental decreases | LP |
| Entropy of two successive letters in equiprobable 3-letter symbolization | EN |
| change in correlation length after iterative differencing | CC |
| exponential fit to successive distances in 2-d embedding space | EF |
| ratio of slower timescale fluctuations that scale with DFA (50% sampling) | FDFA |
| ratio of slower timescale fluctuations that scale with linearly rescaled range fits | FLF |
| trace of covariance of transition matrix between symbols in 3-letter alphabet | TC |
| periodicity measure | PM |

spatial arrangement of the selected electrodes (dark shaded: best electrode; middle shaded: best 5 electrodes; light shaded: best 10 electrodes), and the left-hand side shows the ordered listings (top: electrodes; second from the top: best measures in the group of the single best electrode; third from the top: best measures in the group of the 5 best electrodes; bottom: best measures in the group of the 10 best electrodes).

Selecting specific electrodes (and measures) is a necessary step to proceed to the automatic learning phase; nonetheless, the results of the selection themselves can be interpreted. As it can be observed, the electrodes that seem to encompass the most information are the same in all three binnings, which could be interpreted as an indication that the problem is well-founded. In particular, the best 5 electrodes are always the same (in descending order of e-score: 4L, 6Z, 2LA, 10R, 4LB), while the second 5 best vary a little from binning to binning. Similarly, the 4 measures that occur more often in all selections are Z10, EN, FLF, and FMAI; notably, EN never shows a high variance on electrode 4L, but it emerges as an informative measure as soon as the other (good) electrodes are taken into consideration. As a final observation, we can compare the selected electrodes with those that may be affected by potential noise that has not been cancelled in the preprocessing phase, typically, eye movements [39] and neck muscle contractions [40]. As a result of such a comparison, it is immediate to see that, as expected, the electrodes that have been chosen are not known to be influenced by this kind of noise.

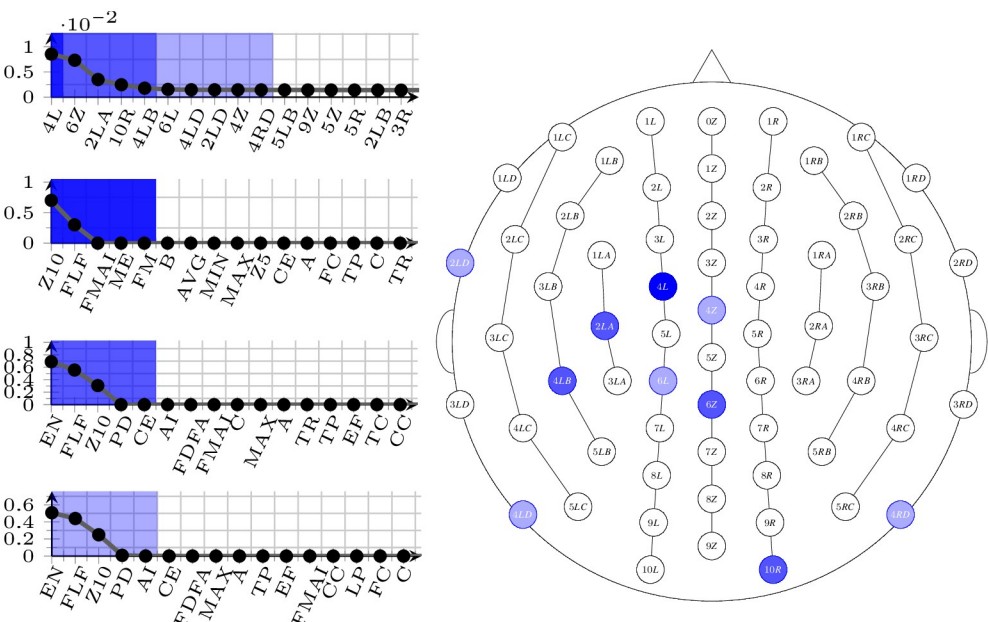

**Fig 5. Selected electrodes and measures for the dataset $D_{25}$ corresponding to the 25-25 binning.** On the right-hand side, their spatial distribution (from dark to light: best one, best five, best ten), and on the left-hand side, top, their ordering. On the left-hand side, also, ordering of the best measures with the different selections (from line 2 to bottom: with the best ten measures, the best five, and with the best electrode only.)

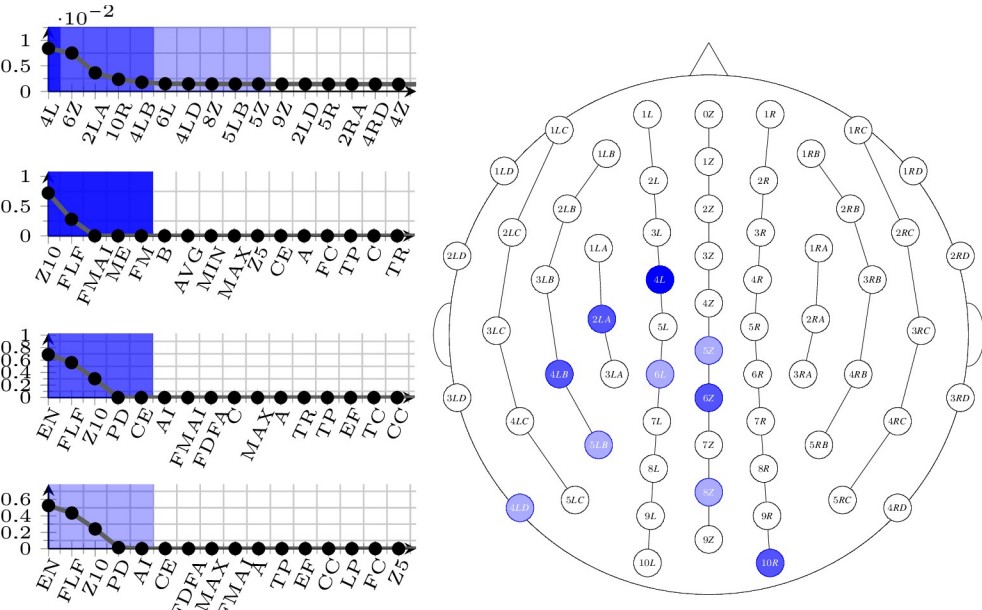

**Fig 6. Selected electrodes and measures for the dataset $D_{34}$ corresponding to the 17-34 binning.** On the right-hand side, their spatial distribution (from dark to light: best one, best five, best ten), and on the left-hand side, top, their ordering. On the left-hand side, also, ordering of the best measures with the different selections (from line 2 to bottom: with the best ten measures, the best five, and with the best electrode only.)

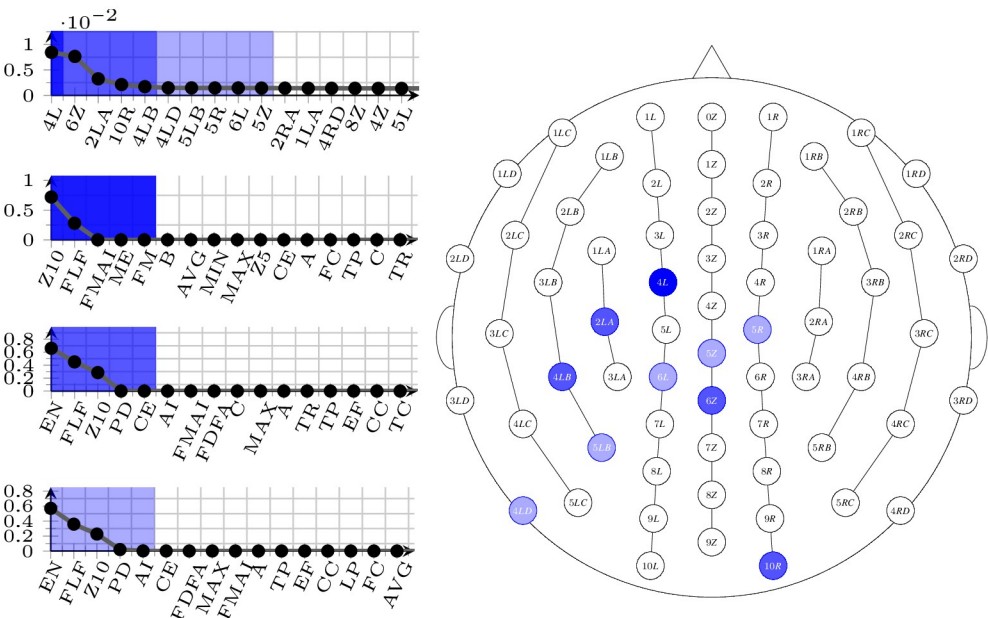

**Fig 7. Selected electrodes and measures for the dataset $D_{41}$ corresponding to the 10-41 binning.** On the right-hand side, their spatial distribution (from dark to light: best one, best five, best ten), and on the left-hand side, top, their ordering. On the left-hand side, also, ordering of the best measures with the different selections (from line 2 to bottom: with the best ten measures, the best five, and with the best electrode only.)

## Learning and classification

Time series classification can be approached in several ways within the context of machine learning. Methods for classifying time series can be roughly separated into *symbolic* and *functional* (see, among others, [41–43]). Symbolic methods aim to extract a logical characterization of the classes in terms of the behaviour of the series, while functional ones approach the classification problem by extracting a mathematical function of the series. Time series classification methods can also be separated into *native* or *feature-based* (see, e.g., [44, 45]). Native methods consider time series as they are, without performing any modification of the signals. Feature-based method, on the other hand, focus on extracting statistically interesting measures of the signals and use those for the classification phase. Feature-based methods are far more common, and they can be symbolic or functional; their major drawback is the lack of interpretability of the results in the functional case, and the low predictive capabilities in the symbolic one. Native methods are scarcer, and the most common ones among them, so-called *distance-based methods* (see [46, 47]), despite their general good behaviour in terms of performances, do not offer a real grasp of the underlying problem.

In [30, 48], a new class of symbolic, native time series classification methods was proposed. Despite their short history, *temporal decision trees*, and their ensemble counterpart *temporal decision forests*, showed a good compromise between interpretability and performances. The key points that define temporal decision trees are:

- They follow the general pattern and schema of conventional decision trees, in which decisions are taken on a dataset in order to maximize the amount of *information gain* in a greedy fashion, starting from the original training dataset and obtaining, at each step, smaller, and more informative subsets. When the dataset associated with a node is too small, or too pure in terms of class, it is converted into a leaf, and labelled with the majority class (generating,

| symbol | Allen's relation | graphical representation |
|--------|------------------|-------------------------|
| $\langle A \rangle$ | $[x,y]R_A[z,t] \Leftrightarrow y = z$ | |
| $\langle L \rangle$ | $[x,y]R_L[z,t] \Leftrightarrow y < z$ | |
| $\langle B \rangle$ | $[x,y]R_B[z,t] \Leftrightarrow x = z, t < y$ | |
| $\langle E \rangle$ | $[x,y]R_E[z,t] \Leftrightarrow y = t, x < z$ | |
| $\langle D \rangle$ | $[x,y]R_D[z,t] \Leftrightarrow x < z, t < y$ | |
| $\langle O \rangle$ | $[x,y]R_O[z,t] \Leftrightarrow x < z < y < t$ | |

**Fig 8. Allen's interval relations and their notation in temporal decision trees.**

as in the classical case, a certain amount of misclassifications). Classical techniques, up to and including *pre- and post-pruning*, can be applied, at least in a limited form, to non-temporal and temporal decision trees alike.

- Unlike conventional decision trees, decisions are relativized to intervals of the time series. So, while conventional decision trees treat times series by extracting features from them, and then taking decisions on such features, temporal decision trees take decisions directly on time series, in a native way. Consider, for example, the average; while a conventional decision tree may separate the dataset using the fact that the average of a specific variable on the whole time period exceeds a given threshold value (e.g. *if the average value of a variable is more than that value, then...*), a temporal decision tree may do so using the existence of an interval in which the average of a specific variable exceeds the same threshold value (e.g. *if the average value of a variable is more than that value between the instants x and y, then...*).

- Like conventional decision trees, a temporal decision tree has a clear logical interpretation, but makes use of a more complex logic than propositional logic, which allows one to express properties over intervals, and their relations. There are thirteen relations between two intervals, known as *Allen's* relations (see Fig 8, in which we show only the six *direct* relations of the type $\langle X \rangle$; their inverses, denoted with $\langle \overline{X} \rangle$, can be obtained by switching the roles of each interval, and the thirteenth, *equals*, is denoted $\langle = \rangle$), and a temporal decision tree is able to learn interval patterns which we can formalize using suitable symbols to denote Allen's relations (see Fig 8, first column).

In [30, 48] it was shown that temporal decision trees perform, in general, better than their non-temporal counterparts, and, while retaining a very high level of interpretability, they are still able to extract classification models that are comparable with those extracted by non-interpretable approaches.

## Experiments and results

For these experiments, temporal decision trees in their *random forest* generalization were used, and the models were trained on the selected electrodes and measures, using all frequency bands $F_1, ..., F_{13}$ (from $\delta$ to $\gamma$), in different combinations. In all experiments, we set the number of trees to 100. Since the three datasets are different in terms of number of instances, in order to ensure comparability of the results we ran all the experiments in the so-called *leave-p-out* cross-validation [49], with $p = 10$; the leave-p-out method consists in training the model leaving $p$ instances as validation set and repeating this process over all the possible

**Table 2. Results of the experiments in the original conditions; all values are expressed in percentage points.**

| | | nelectrodes = 1 | | | | nelectrodes = 5 | | | | nelectrodes = 10 | | | |
|---|---|---|---|---|---|---|---|---|---|---|---|---|---|
| | | *acc* | *avg − acc* | *sens* | *spec* | *acc* | *avg − acc* | *sens* | *spec* | *acc* | *avg − acc* | *sens* | *spec* |
| $\mathcal{D}_{25}$ | *all* | 52 ± 19 | 53 ± 19 | 37 ± 21 | 66 ± 26 | 63 ± 16 | 62 ± 19 | 48 ± 22 | 76 ± 21 | 66 ± 11 | 68 ± 10 | 68 ± 17 | 69 ± 17 |
| | *β* | **70 ± 15** | 67 ± 17 | 53 ± 24 | 82 ± 14 | 56 ± 15 | 55 ± 16 | 54 ± 18 | 56 ± 25 | 60 ± 17 | 59 ± 18 | 52 ± 26 | 66 ± 19 |
| | *γ* | 59 ± 15 | 58 ± 17 | 46 ± 26 | 71 ± 15 | 64 ± 14 | 64 ± 16 | 60 ± 25 | 68 ± 19 | 56 ± 13 | 57 ± 15 | 49 ± 18 | 65 ± 19 |
| | *β + γ* | 63 ± 20 | 65 ± 20 | 50 ± 23 | 79 ± 30 | 60 ± 14 | 61 ± 14 | 56 ± 21 | 65 ± 21 | 66 ± 17 | 65 ± 20 | 58 ± 33 | 72 ± 19 |
| $\mathcal{D}_{34}$ | *all* | 58 ± 12 | 62 ± 12 | 44 ± 17 | 80 ± 19 | 63 ± 19 | 62 ± 18 | 51 ± 28 | 72 ± 19 | 59 ± 17 | 61 ± 19 | 50 ± 30 | 71 ± 12 |
| | *β* | 66 ± 14 | 66 ± 14 | 53 ± 16 | 80 ± 20 | 65 ± 12 | 67 ± 10 | 51 ± 15 | 83 ± 13 | 65 ± 16 | 67 ± 16 | 55 ± 20 | 80 ± 19 |
| | *γ* | 67 ± 13 | 68 ± 9 | 56 ± 23 | 80 ± 18 | 62 ± 10 | 61 ± 8 | 53 ± 19 | 70 ± 16 | 60 ± 11 | 60 ± 13 | 64 ± 13 | 56 ± 21 |
| | *β + γ* | 66 ± 10 | 65 ± 10 | 47 ± 18 | 82 ± 20 | 66 ± 17 | 69 ± 15 | 63 ± 24 | 74 ± 22 | 57 ± 16 | 57 ± 16 | 48 ± 23 | 66 ± 20 |
| $\mathcal{D}_{41}$ | *all* | 74 ± 13 | 75 ± 9 | 57 ± 18 | 92 ± 10 | **71 ± 9** | 70 ± 11 | 57 ± 25 | 83 ± 14 | 76 ± 10 | 80 ± 7 | 72 ± 16 | 89 ± 15 |
| | *β* | 76 ± 13 | 82 ± 9 | 64 ± 18 | 100 ± 2 | **71 ± 13** | 73 ± 14 | 64 ± 15 | 83 ± 19 | 77 ± 11 | 79 ± 11 | 69 ± 17 | 89 ± 13 |
| | *γ* | 77 ± 12 | 78 ± 12 | 71 ± 12 | 85 ± 15 | **74 ± 6** | 76 ± 7 | 68 ± 11 | 83 ± 14 | **72 ± 11** | 68 ± 15 | 59 ± 26 | 78 ± 19 |
| | *β + γ* | **78 ± 9** | 81 ± 9 | 70 ± 14 | 92 ± 10 | 69 ± 13 | 68 ± 12 | 61 ± 17 | 75 ± 12 | **78 ± 11** | 78 ± 11 | 66 ± 17 | 89 ± 12 |

combination of *p* instances with different random seeds. This cross-validation method is mostly adopted when dealing with small datasets; to maintain consistency across our experiments we limited ourselves to 10 repetions out of all possible combinations; leave-p-out generalizes the more standard cross-validation method and ensure the comparability of results between datasets of very different cardinalities, and choosing *p* = 10 is mostly accepted in the literature in terms of number of folds [50].

For each of the three datasets, we explored 4 different configurations of frequency bands, *(i)* all the 13 frequency bands *(ii)* only those corresponding to *β*, *(iii)* only those corresponding to *γ* and *(iv)* only those corresponding to *β* and *γ*. From each of the signals recorded by the selected electrodes, interval temporal random forests were trained on the 5 best measures; we performed three different groups of experiments, using the first 10, 5 and the single best electrode(s), as explained in the previous section. As a consequence, we have a total of 36 experiments composed of 10 seeds each. In Table 2, we show a summary of the validation performances; for each experiment, the table reports the average and standard deviation of four performance metrics, namely, the overall accuracy (the fraction of correctly classified instances), the average accuracy (the average between sensitivity and specificity), the sensitivity (the fraction of likes that were correctly classified), and the specificity (the fraction of dislikes that were correctly classified); our datasets not being perfectly balanced makes the average accuracy slightly more informative than the overall accuracy, the latter being, however, a more standard metric, usually preferred in terms of comparison among different experiments. Temporal decision trees and random forests, as their non-temporal counterparts, are knowingly *greedy* algorithms that return, in general, sub-optimal models; to ensure a deeper exploration of the solution space, we also run all experiments in the same conditions, but selecting only the best measure (instead of the best 5 ones); the results of this version of the experiments are shown in Table 3.

Upon observing Table 2, a few observations can be made. Varying the dataset from $\mathcal{D}_{25}$ to $\mathcal{D}_{34}$, and then to $\mathcal{D}_{41}$, one observes how the validation accuracy tends to improve, which shows that, as expected, training a model with clearer like/dislike instances (that is, instances with more extreme grades) improves the ability of the model itself to pick up the underlying temporal patterns; the same happens when varying the number of electrodes from 1 to 5, and, then, to 10, although not as clearly. Another general observation is that all the trained models tend

**Table 3. Results of the experiments after selecting only the best measure; all values are expressed in percentage points.**

| | | nelectrodes = 1 | | | | nelectrodes = 5 | | | | nelectrodes = 10 | | | |
|---|---|---|---|---|---|---|---|---|---|---|---|---|---|
| | | acc | avg − acc | sens | spec | acc | avg − acc | sens | spec | acc | avg − acc | sens | spec |
| $\mathcal{D}_{25}$ | all | 53 ± 21 | 55 ± 20 | 39 ± 23 | 68 ± 27 | 59 ± 9 | 58 ± 11 | 48 ± 19 | 68 ± 19 | 57 ± 13 | 61 ± 13 | 58 ± 22 | 64 ± 23 |
| | β | **70 ± 14** | 67 ± 17 | 54 ± 25 | 80 ± 14 | 57 ± 19 | 55 ± 20 | 52 ± 24 | 59 ± 28 | 53 ± 14 | 53 ± 16 | 47 ± 25 | 60 ± 19 |
| | γ | 61 ± 16 | 60 ± 17 | 46 ± 25 | 73 ± 17 | 64 ± 16 | 64 ± 19 | 59 ± 31 | 69 ± 24 | 54 ± 11 | 55 ± 13 | 49 ± 18 | 61 ± 17 |
| | β + γ | 66 ± 22 | 66 ± 21 | 54 ± 26 | 78 ± 30 | 58 ± 14 | 58 ± 13 | 55 ± 16 | 61 ± 21 | 62 ± 16 | 62 ± 20 | 52 ± 37 | 72 ± 20 |
| $\mathcal{D}_{34}$ | all | 61 ± 12 | 64 ± 11 | 45 ± 18 | 83 ± 16 | 64 ± 23 | 65 ± 23 | 57 ± 31 | 73 ± 22 | 61 ± 18 | 62 ± 19 | 57 ± 21 | 67 ± 22 |
| | β | 67 ± 14 | 68 ± 15 | 52 ± 16 | 85 ± 17 | 66 ± 15 | 67 ± 14 | 54 ± 17 | 80 ± 14 | 68 ± 14 | 70 ± 14 | 60 ± 15 | 80 ± 18 |
| | γ | 66 ± 13 | 67 ± 9 | 54 ± 24 | 80 ± 13 | 62 ± 9 | 61 ± 8 | 53 ± 15 | 68 ± 21 | 64 ± 12 | 65 ± 12 | 64 ± 19 | 67 ± 14 |
| | β + γ | 66 ± 9 | 64 ± 9 | 48 ± 19 | 80 ± 19 | 67 ± 16 | 68 ± 13 | 64 ± 23 | 73 ± 14 | 63 ± 17 | 63 ± 16 | 54 ± 26 | 72 ± 15 |
| $\mathcal{D}_{41}$ | all | 73 ± 15 | 73 ± 12 | 57 ± 18 | 90 ± 11 | **76 ± 8** | 74 ± 12 | 69 ± 28 | 79 ± 16 | 77 ± 10 | 82 ± 6 | 72 ± 15 | 92 ± 15 |
| | β | 76 ± 13 | 82 ± 9 | 64 ± 18 | 100 ± 0 | 77 ± 13 | 78 ± 14 | 77 ± 15 | 80 ± 19 | 77 ± 13 | 78 ± 14 | 71 ± 21 | 85 ± 13 |
| | γ | 75 ± 11 | 74 ± 11 | 71 ± 12 | 77 ± 17 | **79 ± 7** | 79 ± 9 | 78 ± 19 | 81 ± 14 | **76 ± 10** | 73 ± 15 | 67 ± 25 | 78 ± 22 |
| | β + γ | 76 ± 11 | 79 ± 12 | **70 ± 14** | 88 ± 16 | 70 ± 14 | 67 ± 13 | 64 ± 19 | 70 ± 18 | **81 ± 12** | 81 ± 13 | 75 ± 16 | 86 ± 15 |

to be more specific than sensitive, that is, the models have a greater capacity to distinguish a dislike than a like. Moreover, focusing on the frequencies β and γ (and their combination) tends to improve the performance, with respect to when all frequencies are used. Thus, the best result in terms of average accuracy corresponds to the case of the model learned with the frequencies within β + γ on $\mathcal{D}_{41}$, with a value of 78%, with a specificity of 89%, which means that it correctly recognizes almost 9 dislikes out of 10 while retaining a sensitivity of 66%. We can also observe how the difference between $\mathcal{D}_{25}$ and $\mathcal{D}_{34}$ tends to be small, which may suggest how the subjects are unclear in their own judgments when the judgment itself is not extreme. Perhaps unexpectedly, using only one measure instead of five has a positive effect on the performances of the models, as it can be seen in Table 3. While all the above considerations remain valid, one observes that the models extracted from $\mathcal{D}_{41}$, using 10 electrodes, show very good performances across all frequency bands, and in the case of β + γ it reaches an average accuracy of 81%, with a good balance between specificity (86%) and sensitivity (75%). As we have seen, only a very few works in the recent literature addressed the problem of identifying the areas of the brain, and the frequencies, in which a certain experience seems to manifest itself; in most cases the electrodes are chosen on the basis of the previous neurophysiological literature, and the signals are separated in well-known bands, as opposed to our approach in which important electrodes and measures are identified from the data, and frequencies are explored with finer granularity. Concerning the accuracy with which the liking experience can be recognized by a model, it is, again, very hard to compare the results of very different experiments; it could be said, however, that our results (up to 81% of average accuracy, as we have seen) are very much comparable with the accuracies that emerges across the literature that mentions addressing liking directly (that is, 60%—86%), with two important *caveat*s: our experiment was carried on in a real environment, and our model is interpretable. As a final experiment, we considered $\mathcal{D}_{41}$, in the cases 10, 5, and 1 electrode(s), with the best measure only, and we learned a classification model from such data after having shuffled the labels; we run such process 5 times with 5 different random seeds. As expected, the accuracies have lowered and plateaued around 45%, which, in a balanced setting, corresponds to a random model with zero extracted knowledge. This confirms the robustness of our previous results.

Extracting patterns from biosignals, such as an EEG, is a task that belongs to the machine learning domain. As we have already recalled, however, in EEG and MRI pattern and

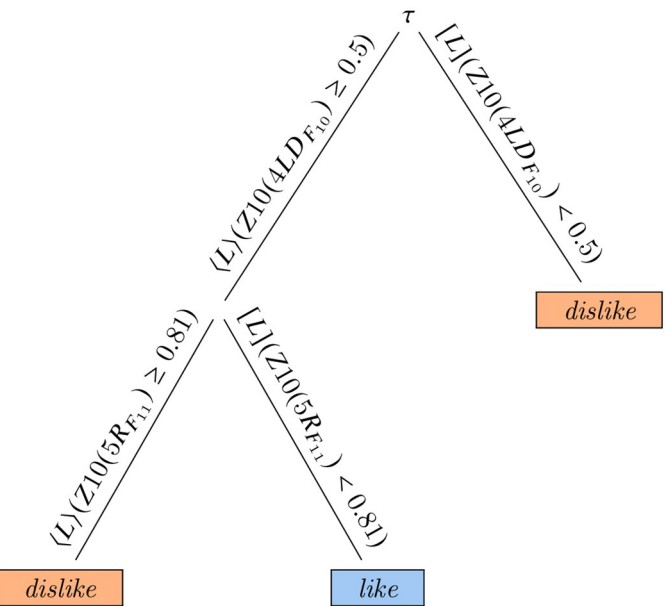

**Fig 9. A temporal decision tree from the model extracted with the best 5 measures.**

knowledge extraction the recent literature focused on functional machine learning techniques, and, in most cases, neural networks. The most relevant drawback of a functional learning schema is that the extracted model can only be evaluated from a statistical point of view. Classic symbolic learning techniques, such as decision trees or rule-based classification has never been a serious alternative for this particular task, due to their lack of expressive power. As it happened in other, very different contexts such as COVID-19 diagnosis [48] and land cover classification [51], temporal (and spatial) decision trees and forests fill, to some extent, this gap, being able to extract complex, but explicit patterns from data. Building on the same idea, therefore, we can now extract explicit trees from the models obtained as a result of our experiment, and discuss them from the point of view of working toward understanding how the EEG signal behaves in subjects in our case. The tree shown in Fig 9 has been learned from $\mathcal{D}_{41}$, using 10 electrodes, the bands in the $\gamma$ spectrum only, and the best 5 measures; its random forest counterpart, as per Table 2, shows 72% of accuracy, on average. This particular tree has, however, 100% validation accuracy, and it reaches such a performance with only five nodes, of which, three are leaves. From it, one can easily extract two rules:

$$\begin{cases} \langle L \rangle (Z10(4LD_{F_{10}}) \geq 0.5 \wedge [L](Z10(5R_{F_{11}}) < 0.81 & \Rightarrow & like \\ [L](Z10(4LD_{F_{10}}) < 0.5 \vee \langle L \rangle (Z10(5R_{F_{11}}) \geq 0.81 & \Rightarrow & dislike. \end{cases}$$

Although interpreting such a rule system from a neurobiological point of view may be misleading, one can observe that, as per this model, the sensation of like is witnessed by the presence of an interval of time (sometimes during the observation) in which the most frequent value of amplitude of voltage at the frequency $F_{10}$, having discretized the possible values in 10 bins, is greater than 0.5 in the electrode $4LD$, while it is never the case that such value is greater than 0.81 in the electrode $5R$ at the frequency $F_{11}$. In the same way the model seems to suggest that if the most frequent value of amplitude of voltage at the frequency $F_{10}$ in the electrode $4LD$ is always less than 0.5 or it reaches a peak greater than 0.81 in the electrode $5R$ at the

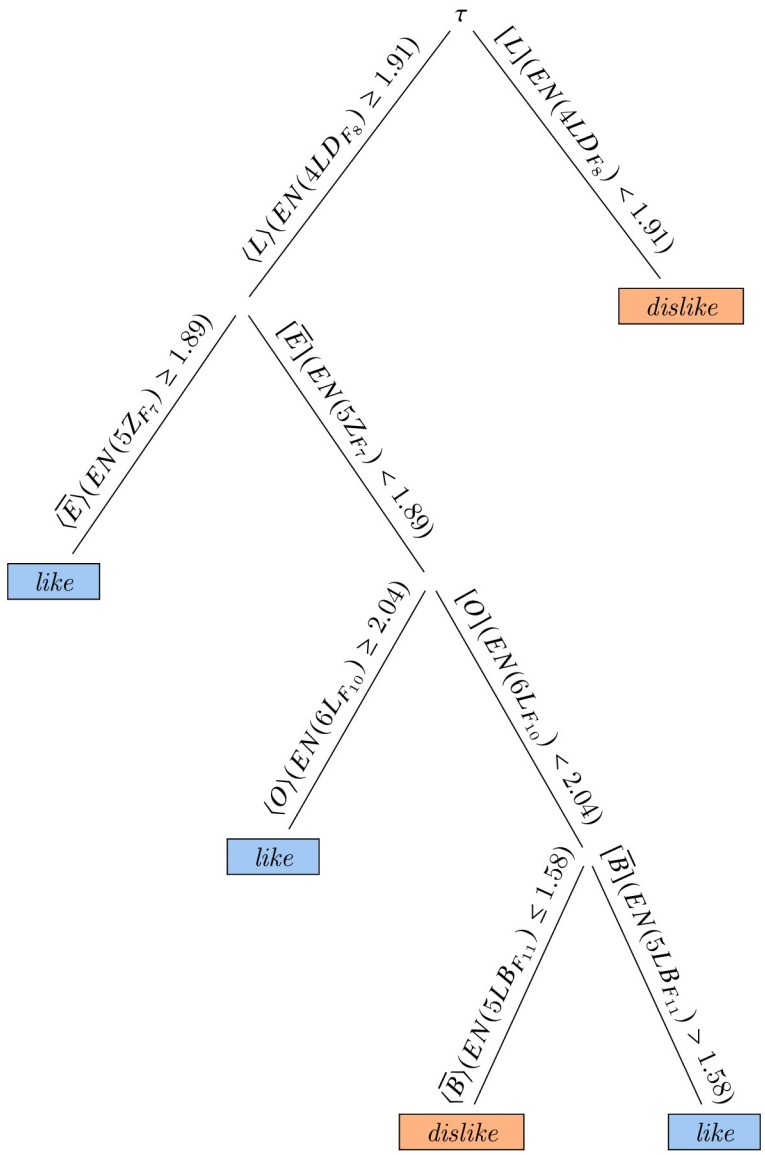

**Fig 10. A temporal decision tree from the model extracted with the single best measure.**

frequency $F_{11}$ sometimes during the observation, the subject should be experimenting a feeling of dislike. As one may see, explicit models such as those extracted by temporal decision trees and forests are qualitative in temporal terms (for example there is no indication on the length of such spikes) but quantitative in event terms. Such an analysis is emblematic of the difference between functional and symbolic models; in the former case, the statistical value is the only indication of the performance of a model, while in the latter case models can be inspected. Even in an ideal situation such as the one under analysis, with 100% validation accuracy, however, precise rules such as the above ones should be taken with precaution, and the experiments should be repeated in different conditions. For comparison, a similar exercise can be done for models extracted using only the best measure. In particular, in Fig 10, we show a tree learned, again, from $\mathcal{D}_{41}$ using 10 electrodes; this time, however, we have chosen a tree learned using bands in the $\beta + \gamma$ spectrum, to be compared, therefore, with a random forest model with

81% of accuracy. The tree alone shows, once more, 100% validation accuracy, but, this time, patterns are temporally more complex; the most important electrode is still *4LD*, over which the first decision is taken, while the other are, *5Z*, *6L*, and *5LB*.

It should be noticed, finally, that using symbolic models such as temporal decision trees has an unparalleled computation advantage. As a matter of fact, after the (offline) learning phase, models are, as shown above, simply sets of rules. As such, they can be implemented in a very simple way. All pre-processing steps, including Fourier transforms, can be efficiently implemented, which contributes to obtain a potentially online tool that provides a real-time, nearly-instantaneous classification. Our post-hoc analysis, specifically, the after-shuffling model learning, shows that, as expected, our models seem to be robust against potential noises of a real environment, such as eye movements and neck muscle contractions.

## Conclusions

In this paper we applied a new methodology to extract knowledge from EEG signals, in order to study a problem of neuroaesthetics. In a sense, we tried to investigate and make clear a possible correlation between EEG signal and the experience of liking an art painting, within the limits of technical appliances. Our data consisted of the EEG recordings of subjects exposed to art paintings, and we analysed them to extract rules that allows one to establish if the subject liked, or disliked, the painting he/she was seeing. The more general problem of emotion recognition from the EEG recording of a subject has been widely debated in the recent literature. While our results must be considered only preliminary, they constitute a first step towards using a new generation of knowledge extraction methods, by means of which one trades, in a limited way, some degree of performance of a prediction model (i.e., reliability of the prediction) to obtain, again in a limited way, explicit knowledge. Our results, however, show performances that are comparable with the existing literature, and our methodology is able to provide, in a systematic fashion, several useful suggestions towards data understanding, including the importance of every electrode, the relevance of the exact frequency at which the information is carried, and the most informative measures that should be applied to the signal for such information to emerge. If, on the one side, comparing our results with existing ones may not be very informative, considering the very different underlying conditions, on the other side they can be seen as one step forward towards synthesizing a theory of subjective liking. While modal symbolic learning is still at its infancy, it is showing the ability to pick up complex patterns from data of very different origin; applying it, in its temporal version, to bio-signals such as EEG, allows one to deal with large amount of data that emerge from a limited amount of trials, which is not only the most typical situation in neurophysiology, but it is also the most effective way to extract all possible information from the data themselves.

In terms of future work, observe that this extraction and analysis technique for EEG signals, applied here in the neuroaesthetic domain, may be feasible for other clinical research topics, for example, to evaluate the functional response and efficacy to pharmacological therapies or to carry out a prospective analysis of patients with cognitive deficits or for monitoring alterations of consciousness, both for diagnostic and prognostic purposes, in states of coma, vegetative state, and/or with minimally conscious state (see, e.g., [52]), in which it is important to have a diagnostic picture and a prognostic evaluation as reliable as possible. Moreover, the idea of employing an algorithm that interprets the like/dislike level of a subject when observing artwork may be used to explore further issues by other researchers in different disciplines. For example, it could be applied in the field of the so-called *everyday aesthetics* (e.g., observing a landscape, an animal, an architectural object), and, even more, in epistemological cognitive disciplines (e.g., psychiatry, psychology), and, more in general, in the biological/physiosophical

sphere in which the *sense of beauty* falls, given that aesthetic pleasure is the very essence of knowledge, a bridge between visual perception and subjective/objective experience of the outside world (according to the theory of structural coupling of Maturana and Varela [53]).

## Acknowledgments

We acknowledge the support of the authorities of the Ferrara Municipality as well as all the staff of Estense Castle and Musei of Arte Antica, Ferrara, in particular Dr. Ethel Guidi and Dr. Giovanni Sassu, for hosting NEVArt at the exhibition *Painting affections: sacred painting in Ferrara between the '500 and the '700*", and for supporting the organization of this research. We also acknowledge the help of Pietro Avanzini, Giovanni Vecchiato e Maddalena Fabbri Destro, researchers at the Neuroscience Institute at CNR (Parma, Italy), for the precious support in both the research design and the tools choice. Finally, a special thanks to the many volunteers who participated directly in the development of the research and to the teachers of the L.Ariosto Classical High School of Ferrara—Anna Maria Masi, Caterina Pieri, Stefania Borini and Francesca Papaleo—and to their students for their help during data collection.

## Author Contributions

**Conceptualization:** S. Mazzacane.

**Data curation:** G. Sciavicco.

**Funding acquisition:** S. Mazzacane, M. Coccagna.

**Investigation:** S. Mazzacane, M. Coccagna, F. Manzella, G. Pagliarini, E. Caselli, G. Sciavicco.

**Methodology:** M. Coccagna, E. Caselli, G. Sciavicco.

**Software:** F. Manzella, G. Pagliarini.

**Visualization:** F. Manzella, G. Pagliarini.

**Writing – original draft:** M. Coccagna, F. Manzella, G. Pagliarini, G. Sciavicco.

**Writing – review & editing:** S. Mazzacane, M. Coccagna, V. A. Sironi, A. Gatti.

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
