## [Decision Letter · Decision Letter 0]

17 Jan 2023

PONE-D-22-26500Towards Measuring the Sense of BeautyPLOS ONE

Dear Dr. Sciavicco,

Thank you for submitting your manuscript to PLOS ONE. After careful consideration, we feel that it has merit but does not fully meet PLOS ONE’s publication criteria as it currently stands. Therefore, we invite you to submit a revised version of the manuscript that addresses the points raised during the review process. If you decide to submit a revised version of the manuscript, I invite you to follow carefully both Reviewers' advice.

Furthermore, I recommend the Authors to address these additional specific points:

1) The Authors have to carefully specify how the present manuscript differs from this previous publication: https://link.springer.com/chapter/10.1007/978-3-031-06242-1_53

If the Authors do not demonstrate that the two manuscripts differ significantly, I will not be able to accept the present manuscritp for publication. At the moment, it seems to me that the two manuscritp show at least a partial text overlap, which is not acceptable. The Authors should be very careful in explaining why the two manuscritps are different, consideranding data, analyses, results and main text.

2) The Authors should state how they selected their sample size, with appropriate explanations, and possibly adding a power analysis, if they performed one.

3) Because of Plos One policy, Authors are required to make their data publicy available.

4) The Authors should expand their discussion section, trying to further discuss their results also highlithing shared aspects and possible discrepancies with previous neuroaesthetic evidence. Also it seems to me that Authors' review of previous EEG literature on neuroaesthetic is a bit scarce. Finally I fully agree with Reviewer 2 when they expressed the importance of distinguishing between beauty and liking. I think this is a crucial theretical point which needs to be addressed in the review of the manuscritp. I think that in light of this Reviewer advice, the Authors should reconsider whether to modify the title.

We look forward to receiving your revised manuscript.

Kind regards,

Irene Ronga, Ph.D.

Academic Editor

PLOS ONE

Journal Requirements:

4. Please include a caption for figure 1.

Reviewers' comments:

Reviewer's Responses to Questions

**Comments to the Author**

1. Is the manuscript technically sound, and do the data support the conclusions?

Reviewer #1: Yes

Reviewer #2: Yes

2. Has the statistical analysis been performed appropriately and rigorously? 

Reviewer #1: Yes

Reviewer #2: I Don't Know

3. Have the authors made all data underlying the findings in their manuscript fully available?

Reviewer #1: No

Reviewer #2: No

4. Is the manuscript presented in an intelligible fashion and written in standard English?

Reviewer #1: Yes

Reviewer #2: Yes

5. Review Comments to the Author

Reviewer #1: General comments

The paper addresses with a novel methodology the traditional concern of neuroaesthetics research since its inception: to clarify the correlation between neurophysiological measures and aesthetic evaluation. To that end, it presents EEG data collected in a broader real-world experiment conducted in an art gallery. The authors analyse an EEG dataset of 248 trials from 16 subjects exposed to a total of 54 paintings. EEG data are analysed using a novel symbolic machine learning technique based on temporal decision trees specifically designed to extract information from temporal data. This technique is used to some effect to identify: 1. the sub-group of electrodes whose signal best correlates with the aesthetic liking, 2. the most relevant sub-group of frequencies in which liking seems to occur, and 3. the most relevant sub-group of temporal features of the EEG signal that allow to highlight such information. As a result, the authors extract several symbolic rules that relate the voltage at specific frequencies and in specific electrodes to the subjects' like/dislike judgements.

The paper has several strengths. I welcome the author’s choice of an ecologically valid setting. I also appreciate the authors’ caution in presenting the ambitions and results of their study, as well as their recognition of the limits of their research in a particularly delicate field such as aesthetic inquiry (recognition that runs against the colonial and overly reductionistic attitude of many strands of research in neuroaesthetics). For me, however, the greatest strength and point of interest of the paper is its methodology, which is, to my knowledge, new in this area of research and potentially promising. I like the idea of using this specific knowledge extraction method instead of functional machine learning techniques to trade “some degree of performance of a prediction model… to obtain… explicit knowledge” (even if it must be recognised, as the authors partially do, that performance is not that impressive and that the explicit knowledge obtained is of limited applicability, as it refers only to the single subject expressing a judgement on a single painting). I also agree with the authors that their methodology could provide “several useful suggestions towards data understanding, including the importance of every electrode, the relevance of the exact frequency at which the information is carried, and the most informative measures that should be applied to the signal for such information to emerge.” For this reason, I think the paper could be a valid addition to the literature on electrophysiological measures and aesthetic preferences.

There are however some issues that need to be addressed before the paper deserves publication. Below are some comments on several parts/aspects of the paper, together with some suggestions for improvement (all of which by no means binding). The main concerns I have are about the “Introduction”, the way results are presented in the “Experiments and results” and “Conclusions” sections, and (potentially) the author’s data sharing policy.

Introduction and literature review

In their “Introduction”, the authors distinguish between two kinds of approaches in neuroaesthetics. There are, they say, “top-down processes, in which the essence of beauty undergoes an axiomatic treatment in which the subjective feeling is broken down in its constituting elements, and bottom-up ones, in which some kind of objective data is analysed and related to the subjective expression of beauty”. The distinction between these two approaches, however, does not seem to emerge quite clearly in their description. The authors say that “while the former [top-down approach], e.g. as in [2], is certainly fascinating from an epistemological point of view (it tries to answer the question of whether beauty can be defined), the latter [bottom-up approach] has the advantage of being based on real data, of being systematic, and of using modern analysis techniques”. But arguably also the study that the authors present as an example of the top-down approach is “based on real data”, “systematic” and uses “modern analysis techniques”; moreover, “bottom-up” studies of the kind that the authors then describe arguably also try to “answer the question of whether beauty can be defined”. The references to Sidhu et al. 2018 and to the Weber-Fechner law do not help much either. Perhaps it could be good to clarify a bit more what the distinction between these two approaches exactly amounts to (perhaps with some clearer examples) and the role (if any) that the distinction plays in the authors’ research.

I’ll add about this introduction that the rationale for mentioning certain specific studies instead of others in the literature review is not always clear (especially in the part devoted to MRI studies) The studies mentioned vary a lot in methodology, hypotheses and results and the underlying common ground for mentioning them is not always apparent (apart from the fact that they are all, e.g., “MRI studies on aesthetic preferences” or “studies using computational methods”: but then why mentioning these specific ones among many?). The authors should perhaps clarify a bit more the relevance of the specific studies they mention for their study.

Data collection and data analysis

Sample sizes (16) and number of trials (248) seem adequate given other EEG studies in this area. The data collection procedure seems as rigorous as the ecologically realistic setting allowed. I just have one question in this regard. The authors say that “the experiment included two different ways to collect EEG data, to analyse benefit and boundaries of different tools, procedures and final results. However, in this paper we analyse only the outcomes coming from subjects wearing a dry electrode EEG cap WaveguardTMtouch by ANT 128 Neuro in the 64-channel variant during the exhibition.” Can the authors motivate (in their reply to these comments or in the paper) the exclusion of the data obtained with the other procedure?

The preprocessing of EEG data seems OK from my non-expert vantage point. Perhaps some slight troubles might come from the prepocessing of the data on liking, which were not normally distributed (see p. 6). It is not entirely clear to me why the authors binned the data the specific way they did, so maybe a few words could be added on that.

The process by which the authors selected the most interesting electrodes and measures seems rigorous enough (but take this with a pinch of salt, as I am not an expert on the treatment of temporal features of EEG signal).

As for the experiments with temporal decision trees, I am just a little bit unsure (for lack of expertise) about the authors’ use of the “leave-p-out method”, particularly regarding their choice to limit themselves to 10 repetitions out of all possible combinations (see p. 11). Maybe in this case too the authors’ choice could better justified?

Results and conclusions

Some of the results of the experiments with temporal decision trees as reported in the body of the text at p. 12 seems to differ from those reported in the Tables 2 and 3 at the same page. For example, the authors say that when using both and the models extracted on D41 retain an average sensitivity of 78%, but the table suggests instead a value of 66%. Similarly, the authors suggest that the models extracted from D41 using 10 electrodes show a sensitivity of 81%, but the table suggests a more modest 75%. A similar thing happens at p. 13, where the authors suggest that the random forest that has been generated from D41 using 10 electrodes, the bands in the spectrum only, and the best 5 measures shows 72% of accuracy: but the value in the table is instead 68%. I might be misreading the table in some way or misinterpreting the text. But if not, the figures in the text need revision, and the results might be slightly less impressive than what the in-text discussion suggests.

As for the “Conclusions” section, I would have liked to see the method and results of the study placed more resolutely in the context of the existing literature, particularly the literature mentioned in the “Introduction”. How do the models used by the authors advance neuroaesthetics research? How do they score with respect to other computational models of aesthetic preferences (e.g. those based on various stimulus features)? How exactly these new methods of knowledge extraction contribute to the stated aim of contributing to “synthesising a theory of beauty”? What are the directions for future research in neuroaesthetics opened up by this study? The authors do say that “our results… show performances that are comparable with the existing literature”, but they do not clarify what existing literature exactly they are thinking about. Some references and further discussion here would be helpful.

Language and writing

Language is generally good and adequate to academic standards. There is however a tendency towards wordiness and overly long sentences (see e.g. the second sentence of the “Introduction”), especially in the “Introduction”. There are also a few typos and grammar mistakes scattered throughout the document (see e.g. lines 18, 34, 43, 93, 159, 226, 234).

Figures

Figures are generally clear, of the right format and resolution, and illustrative. Some figures could benefit from a slightly clearer description (e.g., in Figure 2 it is not immediately clear what is highlighted in what colour; Figure 3 could indicate more clearly which frequency band is represented in which line).

Data sharing

Data sharing in this paper does not seem to comply with PLOS ONE’s guidelines. In their submission form, the authors say that “data cannot be shared publicly. Partial sharing may be possible upon contacting with the authors”. However, the journal’s guidelines require “all data underlying the findings to be made available without restrictions”, and clearly specify that “stating ‘data available on request from the author’ is not sufficient”. One needs to provide additional justification for not making data available, which the authors have failed to do. The authors should probably clarify the rationale behind their data sharing choices.

Reviewer #2: See attached file for clear a clear version of the review.

Short summary.

The authors' main aim is to understand the link between the amplitudes of voltage at predetermined frequencies on the sensor level of the EEG signals, and the self-report of beauty while standing in front of several paintings. They are motivated by the fact that prior models offer poor interpretability compared to temporal decision trees (TDTs). As a TDT allows extracting specific rules about the signal and reported outcomes, it is motivated as a valuable model with easily interpretable measures.

Review.

I generally find the symbolic approach to be an underestimated data-driven classification method. As the authors display, it clearly holds great potential in revealing features in the data that may not be picked up by functional approaches. Even more, using more ecologically valid scenarios is exactly the right development for neuroaesthetics being an interesting discipline that has only just begun realizing the importance of context. The authors are certainly on the right track.

There is an increasing trend in applying decision trees to time series, so this is a timely contribution (pun intended). As a TDT allows extracting probabilistic rules, their practical value is evident: if a temporal system is making rule-based decisions, a TDT will surely uncover the rules. However, two questions arise: 1. The uncovering of the rules depends on the quality of the time series, so why is it assumed to be sufficient with EEG signals alone? 2. Why should we consider the brain to exhibit rule-based decisions in liking judgment? Perhaps the authors are not interested in uncovering the relationship between neuronal activity and liking, which is critical in neuroaesthetics. Perhaps the authors are interested in applying a TDT to a neuroaesthetic case. If so, the quality of the data is critical.

While numerous measures are captured, the authors end up using only (dry electrode) EEG signals. This made me wonder whether the gaze pattern, ECG, and EDA could be useful in classifying the self-reported scores. In fact, if the gaze pattern results in a more accurate and sensitive model than the EEG signals alone, it would suggest that the authors are modelling motion artefacts rather than brain signals, which effectively means that the authors are not applying a TDT model to brain signals, but to a combination of convoluted signals. This is not to say that motion is irrelevant to the scores, but it would change the perspective of the paper.

It is unclear whether this paper is dealing with one or both of the following important issues: to quantify the quality of liking, or to give quality to the observed quantities in the model, namely, the rules. If I understood this correctly, the "interpretation" that the authors aim to regain is of the model, not the cortical activity. If so, the quality of the data is immensely important. Needless to say, data quality determines model quality. However, the authors chose to not clean their EEG data.

Perhaps as a next step, the authors could apply strong cleaning pipelines to their EEG dataset (see the work of Klaus Gramann) and see whether that makes the features in the EEG signal even stronger. I suspect this might be the case. If so, it would greatly improve the paper. With that said, I do think the authors are on the right track; both in terms of applying TDT to brain signals and self-reports and regarding the application of real contexts in neuroaesthetics.

Minor issues.

- Typo on p. 5, line 159. It should be ‘[0, 52]’.

- It is unusual to divide brain signals into equally wide bands of frequencies. It makes sense to divide your dataset for computational purposes, but this is at the cost of interpretation. Why would it be useful to model the frequency range of 44 Hz to 48 Hz? In other words, why not simply divide your signal directly into the five wave patterns?

- Figure 2 and 3: Please add the unit (µV I assume?) and be more concrete about each plot. It's difficult to understand what is what. As they look identical, they are surely not band-pass filtered signals. They should look drastically different from one another if that was the case.

- Figure 4: Please include what the various shades of blue mean. It should not only be in the text.

- Typo on p. 9, line 243. Remove the text ‘fix.’

- On page 11, lines 319-324: When designed as a classification problem, did you not expect your model to improve its accuracy when the data approaches a binary structure?

- On page 12, lines 329-333: It is very unclear from where these numbers are coming. Using Table 3, I fail to identify the said numbers. In fact, dataset D41 (under beta + gamma) never scores higher than 75% in sensitivity. Are the reported numbers correct?

- On page 12, lines 333-335: I find this observation very valuable. This is an interesting quality observed in the model: your participants suggest that when it comes to liking or disliking a painting, they tend to not have a middle ground.

- If possible, a few sentences on why/when TDTs are the better choice over HMMs, for instance, would be valuable to the reader. The fact remains that both HMMs and TDTs can be useful for modelling brain signals and can provide some level of interpretability, but the specific strengths and weaknesses of each model may make one more suitable than the other. Why, in your case, is TDT the better approach? I do want to remind the authors that HMMs are particularly useful for modelling systems with a limited number providing insights into the underlying dynamics of the system through the learned model parameters. In giving quality to the quantity (model parameters vs. decision rules), why are decision rules better in your case? These were the thoughts I had while reading your manuscript. A few sentences on this topic would, in my opinion, improve the quality of the paper.

Major issues.

I identify 3 major issues:

1. It is unclear whether the authors are aiming to demonstrate how TDTs have revealed an important decision rule based on EEG signals, which is valuable to interpreting beauty (quantifying quality)—or, that TDTs provide quantities in the form of rules that can be interpreted as some quality of beauty (giving quality to quantity). The unclarity revolves around the motivation being ‘interpretability’. It is unclear what exactly is meant to be better interpreted from the approach. While reading through the introduction, I was convinced that this paper is about fitting a model that allows us to better interpret the neurophysiological data relative to the subjective experience of beauty. It was not until page 11 that I realized the aim may be the exact opposite, namely, to give quality to quantity, which in itself is an important (methodological and philosophical) task. It is important to realize that these are two different objectives. Quantifying quality is to demonstrate how, for instance, (spatial/temporal features of) neural signal(s) correlate with self-reports, revealing the measurable structures of beauty, whereas giving quality to quantity (qualitifying quantity?) is to assume that the modelled system (the brain or the subjective valuation system, in this case), truly operates as the structure of the model (a decision tree, in this case), which is quite a claim. I believe I got confused about the aim as the introduction is strongly dominated by neuroaesthetics, which is centred around quantifying quality.

o If the authors do mean to interpret beauty, it would be more beneficial to introduce what the observed signals from the EEG might mean both neurophysiologically and subjectively. Clearly, the EEG signal is not directly measuring amygdala activity, which the ‘introduction’ discusses, but then what are you classifying, neurophysiologically speaking? By introducing a theory, the interpretation, i.e. quality of quantity, becomes clearer.

o If the authors do not mean to interpret beauty too, then I do not see why the introduction needs to make bold claims about objective and subjective beauty when the main aim of the paper is not to interpret beauty, strictly speaking. This is effectively putting the paper in unnecessary deep waters.

2. This leads me to another point, namely, what are the authors modelling? Preprocessing is an important stage, particularly for modelling purposes. The important point is that the authors are motivated by the lack of meaningful interpretability of prior work. Neural networks can indeed get incredibly difficult to interpret. However, interpretation of what exactly? They begin by choosing not to clean their EEG data (despite referring to a paper from authors that pioneer the cleaning of EEG data, p. 3, lines 99-102). This is why/where I believe the theory could matter. If this paper is about the application of a TDT to EEG signals, the authors should engage with cleaning the data in the sense that they attempt to dissociate gate-related noise, muscle noise, blinking and other eye-related noise, etc.

o Surely, the authors can claim that they are focused on computational interpretability, but at the cost of becoming a methodological (proof of concept) paper about applying TDTs to EEG signals and self-reports. The paper should thus be presented as such and not a case about ‘Measuring the Sense of Beauty’.

3. The authors are confusing liking with beauty. There is a wealth of studies demonstrating the difference between beauty and liking judgment. See for instance the works of Oshin Vartanian, Martin Skov, Marcos Nadal, Anjan Chatterjee, and Helmuth Leder. The title and introduction need a rewrite to align with your data dimensions and methodology.

6. PLOS authors have the option to publish the peer review history of their article (what does this mean?). If published, this will include your full peer review and any attached files.

Reviewer #1: No

Reviewer #2: No

---

## [Author Response · Author response to Decision Letter 0]

25 Feb 2023

All answers are in the attached file 'answersV1.pdf', uploaded as 'response to reviewers'.

---

## [Decision Letter · Decision Letter 1]

3 Apr 2023

PONE-D-22-26500R1Towards an objective theory of subjective liking: a first step in understanding the sense of beautyPLOS ONE

Dear Dr. Sciavicco,

Thank you for submitting your manuscript to PLOS ONE. After careful consideration, we feel that it has merit but does not fully meet PLOS ONE’s publication criteria as it currently stands. Therefore, we invite you to submit a revised version of the manuscript that addresses the points raised during the review process. I am very grateful to both Reviewers for their comments and I would suggest the Authors to try to address them all: especially, if the Authors decide not to modify the preprocessing of EEG data as suggested by Reviewer2, they should justify their choice in the manuscript. Furthermore, I suggest the Authors to include a wider literature review on neuroaesthetics.

We look forward to receiving your revised manuscript.

Kind regards,

Irene Ronga, Ph.D.

Academic Editor

PLOS ONE

Journal Requirements:

Reviewers' comments:

Reviewer's Responses to Questions

**Comments to the Author**

1. If the authors have adequately addressed your comments raised in a previous round of review and you feel that this manuscript is now acceptable for publication, you may indicate that here to bypass the “Comments to the Author” section, enter your conflict of interest statement in the “Confidential to Editor” section, and submit your "Accept" recommendation.

Reviewer #1: All comments have been addressed

Reviewer #2: (No Response)

2. Is the manuscript technically sound, and do the data support the conclusions?

Reviewer #1: Yes

Reviewer #2: Partly

3. Has the statistical analysis been performed appropriately and rigorously? 

Reviewer #1: Yes

Reviewer #2: Yes

4. Have the authors made all data underlying the findings in their manuscript fully available?

Reviewer #1: Yes

Reviewer #2: No

5. Is the manuscript presented in an intelligible fashion and written in standard English?

Reviewer #1: Yes

Reviewer #2: Yes

6. Review Comments to the Author

Reviewer #1: (No Response)

Reviewer #2: See attached version.

Review of Towards Measuring the Sense of Beauty

Review of the revised version

The authors generally answer all my points to a satisfactory level except for the preprocessing point. My concern is best expressed in the justification made on page 6, lines 220-223. Both eye movement and full-body movement artifacts can vary in frequency, depending on the type of movement. Eye movement is easily picked up by an EEG and, opposite to gait and locomotion, it affects several frequency bands. It is thus, in my opinion, necessary to demonstrate that eye movement and other motion artifacts have been excluded.

- See in particular section “3.2 EEG activity around a saccade” in “Nikolaev A.R., R.N. Meghanathan & C. van Leeuwen. 2016. Combining EEG and eye movement recording in free viewing: Pitfalls and possibilities. Brain Cogn. 107: 55–83.” This paper shows that eye-related activity is mostly expressed in the gamma range.

- See also Figure 7 in “Plöchl M., J. Ossandón & P. König. 2012. Combining EEG and eye tracking: identification, characterization, and correction of eye movement artifacts in electroencephalographic data. Front. Hum. Neurosci. 6:.” This paper also shows that eye-related activity is mostly expressed in the gamma range.

- For the case of micro-saccades (not saccades and free-viewing), which I believe this project will contain many of, see “Engbert R. 2006. Microsaccades: a microcosm for research on oculomotor control, attention, and visual perception. In Visual Perception Martinez-Conde S., Macknik S.L., Martinez L.M., et al., Eds. 177–192. Elsevier.” showing that these occur all the way down to the frequency band of 1-2 Hz.

To be specific, my concern is that the quality of the model may be based on eye-related activity conflated with brain activity. Surely, the quality of the model is also based on neural activity, but my concern is that it is driven by eye-related activity, which is suspiciously expressed in the gamma range too. This concern can be addressed by simply applying an ICA analysis and removing the identified eye-related components. Alternatively, the eye-tracking data can be included and contrasted, however, as the authors suggest, this may be an entirely different project. In addition, one could run two models with separate preprocessing settings and compare the outcomes. This is, of course, beyond the scope of the presented work.

Minor issues

There remain just minor issues that I hope the authors would agree to fix to improve the quality of their paper.

- Line 2: To support your case of distinguishing between ‘liking’ and ‘wanting’ in neuroaesthetics, you may consider: ‘Chatterjee A. & O. Vartanian. 2014. Neuroaesthetics. Trends Cogn. Sci. 18: 370–375.’

- Line 3 to 8: There are many “in which”—are you sure you want that structure in your sentences?

- Line 13: It is clear that you are familiar with the Weber-Fechner law and surely know that is not an attempt to quantify beauty. It is a law describing the relationship between physical stimulus and perceived change. Your sentence comes across as if it is a law suggested to quantify beauty.

- Line 20 to 22: To support your case of using ‘liking’ as an emotion, you may want to consider ‘Berridge K. & P. Winkielman. 2003. What is an unconscious emotion? (The case for unconscious “liking”). Cogn. Emot. 17: 181–211.’

7. PLOS authors have the option to publish the peer review history of their article (what does this mean?). If published, this will include your full peer review and any attached files.

Reviewer #1: No

Reviewer #2: No

---

## [Author Response · Author response to Decision Letter 1]

19 May 2023

Dear editor(s)

The paper has been revised and all answers are included in the 'response to reviewers' document.

Thank you for your effort.

---

## [Decision Letter · Decision Letter 2]

7 Jun 2023

Towards an objective theory of subjective liking: a first step in understanding the sense of beauty

PONE-D-22-26500R2

Dear Dr. Sciavicco,

We’re pleased to inform you that your manuscript has been judged scientifically suitable for publication and will be formally accepted for publication once it meets all outstanding technical requirements.

Kind regards,

Irene Ronga, Ph.D.

Academic Editor

PLOS ONE

Additional Editor Comments (optional):

Reviewers' comments:

Reviewer's Responses to Questions

**Comments to the Author**

1. If the authors have adequately addressed your comments raised in a previous round of review and you feel that this manuscript is now acceptable for publication, you may indicate that here to bypass the “Comments to the Author” section, enter your conflict of interest statement in the “Confidential to Editor” section, and submit your "Accept" recommendation.

Reviewer #1: All comments have been addressed

Reviewer #2: All comments have been addressed

2. Is the manuscript technically sound, and do the data support the conclusions?

Reviewer #1: Yes

Reviewer #2: Yes

3. Has the statistical analysis been performed appropriately and rigorously? 

Reviewer #1: Yes

Reviewer #2: Yes

4. Have the authors made all data underlying the findings in their manuscript fully available?

Reviewer #1: Yes

Reviewer #2: Yes

5. Is the manuscript presented in an intelligible fashion and written in standard English?

Reviewer #1: Yes

Reviewer #2: Yes

6. Review Comments to the Author

Reviewer #1: The authors have addressed my concerns. The literature review could still include more relevant work in neuroaesthetics, but the lack of references in that area is justified by the shift in focus from beauty and aesthetics to emotion recognition from EEG signals made by the authors in their first revision. I think the manuscript is now ready for publication.

Reviewer #2: Although I remain reluctant to the statement that the data is free of muscle and eye-movement activity, the authors justify their reasoning based on existing work. I believe this qualfiies their approach, however, I remain sceptical. It could still be that the model is learning based on eye-movement activity rather than brain. I realize that to rule that out, it requires a comprehensive source-localization analysis that I understand is outside the scope of the paper.

Perhaps as an important note: it is, in fact, possible to perform online/real-time ICA. See the REST toolbox, which is rather popular among mobile EEG researchers.

Other than that, the authors have successfully addressed my points.

7. PLOS authors have the option to publish the peer review history of their article (what does this mean?). If published, this will include your full peer review and any attached files.

Reviewer #1: No

Reviewer #2: No

---

## [Editor Report · Acceptance letter]

14 Jun 2023

PONE-D-22-26500R2 

Towards an objective theory of subjective liking: a first step in understanding the sense of beauty 

Dear Dr. Sciavicco:

I'm pleased to inform you that your manuscript has been deemed suitable for publication in PLOS ONE. Congratulations! Your manuscript is now with our production department. 

Kind regards, 

on behalf of

Dr. Irene Ronga 

Academic Editor

PLOS ONE